# Oral Health Workforce in Africa: A Scarce Resource

**DOI:** 10.3390/ijerph20032328

**Published:** 2023-01-28

**Authors:** Jennifer E. Gallagher, Grazielle C. Mattos Savage, Sarah C. Crummey, Wael Sabbah, Benoit Varenne, Yuka Makino

**Affiliations:** 1Dental Public Health, King’s College London, Faculty of Dentistry, Oral & Craniofacial Sciences, Denmark Hill Campus, London SE5 9RS, UK; 2Dental Office, WHO Oral Health Programme NCD Department, Division of UHC/Communicable and NCDs, World Health Organization, 20 Avenue Appia, Geneva 1211, Switzerland; 3Dental Office, Noncommunicable Diseases Team, WHO Regional Office for Africa, Cité Djoué, Brazzaville P.O. Box 06, Congo

**Keywords:** Africa, oral health, health workforce, dental care, low-income population, developing countries

## Abstract

The World Health Organization (WHO) African Region (AFR) has 47 countries. The aim of this research was to review the oral health workforce (OHWF) comprising dentists, dental assistants and therapists, and dental prosthetic technicians in the AFR. OHWF data from a survey of all 47 member states were triangulated with the National Health Workforce Accounts and population data. Descriptive analysis of workforce trends and densities per 10,000 population from 2000 to 2019 was performed, and perceived workforce challenges/possible solutions were suggested. Linear regression modelling used the Human Development Index (HDI), years of schooling, dental schools, and levels of urbanization as predictors of dentist density. Despite a growth of 63.6% since 2010, the current workforce density of dentists (per 10,000 population) in the AFR remains very low at 0.44, with marked intra-regional inequity (Seychelles, 4.297; South Sudan 0.003). The stock of dentists just exceeds that of dental assistants/therapists (1:0.91). Workforce density of dentists and the OHWF overall was strongly associated with the HDI and mean years of schooling. The dominant perceived challenge was identified as ‘mal-distribution of the workforce (urban/rural)’ and ‘oral health’ being ‘considered low priority’. Action to ‘strengthen oral health policy’ and provide ‘incentives to work in underserved areas’ were considered important solutions in the region. Whilst utilising workforce skill mix contributes to overall capacity, there is a stark deficit of human resources for oral health in the AFR. There is an urgent need to strengthen policy, health, and education systems to expand the OHWF using innovative workforce models to meet the needs of this region and achieve Universal Health Coverage (UHC).

## 1. Introduction

Epidemiological research highlights the burden of untreated oral disease in the World Health Organization (WHO) African Region (AFR) [1,2,3], with about half its population suffering from oral diseases, most notably dental caries, periodontal disease, and tooth loss [4]. Whilst most countries in the AFR have traditionally reported a lower caries risk than middle- and higher-income countries, in line with a traditional diet [1,5], nutritional transition to a Western diet has been linked to an increased prevalence of caries [6], even in the presence of hunger and malnutrition. The consequences of oral and dental disease are particularly acute in the AFR. Preventable fatalities associated with acute dental infections are of specific concern [7].

Additionally, there is increasing prevalence of dental trauma in both primary and secondary dentitions [1], variable levels of oral cancer (lip and oral cavity; age-standardized incidence rate of oral cavity cancer of 1.8 per 100,000 population) [5], and HIV with oral manifestations [5]. Cleft lip and/or palate are among the most frequent congenital disorders [1,2,5]. Furthermore, conditions such as Burkitt’s lymphoma [7] and noma [5] present in certain lower-income areas. 

Most oral and dental conditions are preventable, and/or treatable, particularly early in the disease process and should be identified and managed in primary care settings. Moreover, they share common risk factors with other noncommunicable diseases (NCDs), including diet (sugar), tobacco, alcohol, and poor hygiene [8,9]. Strategically, a common risk factor approach is vitally important, together with ensuring optimal fluoride delivery [10], whilst addressing underlying psychosocial and economic determinants [11]. 

National disease management, however, remains limited and vertical, rather than using integrated cost-effective strategies [1,2,3,5]. Furthermore, oral health has a low priority, as demonstrated by the absence of oral health policies, resources, and technical capacity [8,11]. Thus, an AFR Regional Oral Health Strategy 2016–2025 [8] was endorsed, guiding member states to integrate oral health into NCD-prevention and control in the context of universal health coverage (UHC) [11]. Globally in an historical turning point, the 74 [12], and 75th [13] World Health Assemblies have recognised the immense burden created by oral diseases, especially for low-income settings, including ‘to develop innovative workforce models to respond to population oral health needs’ as a key strategic pillar.

The AFR consist of 47 countries: Algeria, Angola, Benin, Botswana, Burkina Faso, Burundi, Cabo Verde, Cameroon, Central African Republic, Chad, Comoros, Congo, Cote d’Ivoire, Democratic Republic of Congo, Equatorial Guinea, Eritrea, Eswatini, Ethiopia, Gabon, Gambia, Ghana, Guinea, Guinea-Bissau, Kenya, Lesotho, Liberia, Madagascar, Malawi, Mali, Mauritania, Mauritius, Mozambique, Namibia, Niger, Nigeria, Rwanda, Sao Tome and Principe, Senegal, Seychelles, Sierra Leone, South Africa, South Sudan, Togo, Uganda, United Republic of Tanzania, Zambia, and Zimbabwe. The region largely comprises low- and middle-income states [14] and is recognised as the most challenged region globally, with limited health workforce capacity [15]. Therefore, the regional strategy [8], guides member states to take the following actions:

Promote capacity building in oral health promotion and integrated disease prevention and management for oral health professionals and other health and community workers matching the oral health needs of the population;

Develop workforce models for integration of basic oral healthcare within primary healthcare (PHC) [8].

In light of population needs and in support of current strategies [8,9,11,12,13], it is of the utmost importance to profile the oral health workforce (OHWF) in the AFR to inform action. Thus, the aim of this paper is to review the OHWF in the AFR, taking account of wider determinants, exploring contemporary challenges and possible solutions. 

The objectives are:To describe the nature of the OHWF, density, trends, distribution, and education (relevant to training oral healthcare providers).To examine the association between OHWF density, country level development, and urban/rural population distribution.To explore OHWF challenges and possible solutions.

## 2. Materials and Methods

OHWF data were obtained from two sources. First, cross-sectional data from a global OHWF service evaluation survey of member states conducted by the WHO in collaboration with King’s College London, United Kingdom (UK), between January and December 2019. The scope of the survey was informed by the literature and the WHO workforce categories: dentists (2261), dental assistants/therapists (3251), and dental prosthetic technicians (3214) (Appendix A), based on the International Standard Classification of Occupations (ISCO-08) [16]. The survey was co-designed, approved, and conducted through WHO offices and comprised open and closed questions across four domains: capacity, capability and governance, education and training, and workforce challenges/solutions (Appendix B). Second, workforce volume and density per 10,000 population, over time, were extracted from the National Health Workforce Accounts (NHWA) [17] for dentists, dental assistants/therapists, and dental prosthetics/technicians. Density per 10,000 population involves dividing the number of reported professionals by the country’s population [18] (Appendix C).

This integrated study database was created through triangulation of data obtained from the above two sources for the 47 countries of the AFR. The latest reports from member states ranged from 2002 to 2019. Data management was performed using Microsoft Excel 365. Where workforce numbers were similar, the NHWA data were used. Where marked differences existed for the same year, the WHO authors liaised with the relevant member states to identify the most reliable data to include for analysis. 

### 2.1. Dataset 

Data on dentists were available for all AFR countries (100%; n = 47). Data coverage for the rest of the OHWF was lower, with 41 countries (87.2%) for dental assistants/therapists and 42 (89.4%) for dental prosthetics/technicians. Countries with missing data on their OHWF were excluded from analysis. 

Whilst most African countries (n = 39, 83%) updated their dentist numbers during 2018 and 2019, data on dental assistants/therapists and dental prosthetics/technicians, tended to be older and updated less frequently (64% of dental assistant/therapists; 66% dental prosthetics/technicians).

### 2.2. Data Analysis

First, OHWF density, trends, distribution, and education were examined. Descriptive analysis was performed to examine data coverage, workforce densities per 10,000 population, and the human development index (HDI). HDI provides information about a country’s education, health, longevity, and income [19]. 

Second, country-level association between OHWF density, development, and urban/rural population distribution was assessed to review the relationship with the national status and population distribution within a country. 

Third, unadjusted associations between density of dentists and density of the combined OHWF with ‘HDI’, ‘mean years of schooling’, ‘number of dental schools’, and ‘percentage of urban population’ were assessed to predict density of dentists and combined workforce through linear regression models for the countries included in the AFR. Previous studies have shown evidence that similar variables are predictors for NCD incidence [20,21,22] and for health workforce [23]. As densities of dentists and combined OHWF were continuous variables, linear regression was chosen as the most appropriate method of analysis. Its standard equation is demonstrated below:

Y = a + bX, where X is the explanatory variable and Y is the dependent variable was used.

In this model, population data and urbanization rates were obtained from the United Nations [18] and the World Bank [24]. Levels of urbanization are related to the presence of developed infrastructure and opportunities for employment after graduation [5]. The predefined distribution categories of HDI were used to describe each country: low (HDI < 0.5), medium (0.5 ≤ HDI < 0.7), high (0.7 ≤ HDI < 0.8), and very high (HDI ≥ 0.8) [19]. The Statistical Package for Social Sciences version 27 (SPSSv27) was used to perform this analysis. 

Fourth, and finally, qualitative data obtained from the global survey on OHWF challenges and possible solutions were examined by the research team descriptively and thematically [25] for member state perspectives.

## 3. Results

There were 36,222 dentists, 32,783 dental assistants/therapists, and 11,986 dental prosthetics/technicians reported for the AFR in 2019. Average density per 10,000 population was 0.44 for dentists, 0.39 for dental assistants/therapists, and 0.10 for dental prosthetics/technicians. The ratio of dentists per dental assistants/therapist was almost equivalent at 1: 0.91, whilst dentists per dental prosthetics/technician was 1:0.33. Countries with the lowest densities were South Sudan for dentists (0.003), Ethiopia for dental assistants/therapists (0.007), and Burkina Faso and the Democratic Republic of Congo (DRC), for dental prosthetic technicians (both 0.001). The Seychelles at 4.297 dentists per 10,000, 3.776 dental assistants/therapists, and 1.432 dental prosthetic technicians had the highest densities for all groups. 

Whilst 11 countries reported having no dental school, 22 countries reported the presence of 61 schools, with evidence of separate training colleges for other OHWF members. 

Despite an overall growth of 63.6% in the density of dentists from 2010 (0.28 per 10,000) to 2019 (0.44, per 10,000), the density is consistently very low. Over the last decade, only four countries consistently exceeded one dentist per 10,000 population (Algeria, Mauritius, Seychelles, and South Africa) (Figure 1) (Appendix D). Available data suggest a higher rate of growth of dental assistants/therapists (238%) and dental prosthetic technicians (250%), with Nigeria having the highest growth for both groups. 

Workforce densities in relation to HDI, presented in Figure 2 and Figure 3, demonstrate a clear social gradient. The average density of dentists per 10,000 population in countries with low HDI is 0.051 against 0.652 for medium HDI or above, whilst for those of ‘high HDI or above’, it is 1.270. 

Linear regression analysis revealed that country-level HDI scores were associated with the density of dentists (Table 1) and the combined OHWF (Table 2). For each unit increase in HDI, there was an increase in density of dentists of 6.53 and in density of combined OHWF of 12.07. Similarly, average years of schooling also showed a significant, but weaker, association with workforce densities but not with the reported number of dental schools and level of urbanization in this unadjusted model. 

When challenges (Figure 4) and possible solutions (Figure 5) were explored, responding countries (n = 40) identified “*maldistribution of the workforce (urban/ rural)*”, “*oral health considered low priority*”, and “*lack of financial support for oral health workforce training institutions*” as the three biggest challenges. Multiple solutions were supported, led by the need to “*strengthen oral health policy*”, “*improve health workforce data for planning*”, and provide “*workforce incentives to work in underserved areas*”. 

Other reported challenges displayed common themes. First there were concerns posed over the “*lack of leadership at national level for oral health*”. Second, there were issues relating to education and training programmes and numbers, where the “*lack of dental auxiliaries such as therapists and assistants*” was identified and challenges regarding coverage given that there are “*landlocked areas not chosen by dentists*”. Third, lack of funding impacted the ability to provide patient care.

Comparable themes were displayed by 17 countries in highlighting additional solutions. Stronger governance was advocated to support action towards an appropriate OHWF, including *“defining clearly, the roles and responsibilities of the dental auxiliaries”* was advocated. Workforce redistribution required action, recognising that *“dentists and other oral health professionals congregate in cities leaving remote areas underserved”* were highlighted. To deliver this change, there was a great need identified by some countries to establish, extend, or re-establish dental education and training, including *“creating a dental school in the country, training existing staff, continuing training, implementing specialties”*, *“re-opening training schools”*, and enabling *“inclusion of essential oral health care services as part of UHC initiatives.”* Additional factors included enriching the system through *“exchanges with other countries”*, *“developing postgraduate training”*, and undertaking *“planning and operational research”.*

## 4. Discussion

To the knowledge of the authors, this is the first paper to comprehensively examine the contemporary OHWF in the AFR. The findings highlight the critically low levels to serve a region with a rapidly growing population, where workforce densities are strongly associated with the level of development (HDI) in particular. There is, however, evidence of embracing a workforce skill mix in similar proportions to dentists. Reported challenges include workforce maldistribution, exacerbated by a lack of financial support for training and workforce data for planning. Strengthening oral health policy and gaining better workforce data were perceived as important, together with having incentives to work in underserved areas to meet population oral health needs.

The average density of the OHWF for the AFR is 0.780 per 10,000 population (0.44 of which relates to dentists). For a continent where significant population growth is predicted [18], urgent action is needed to solve an imminent oral health crisis and address increasing levels of oral disease [5,8,26]. Europe, in contrast, with less than two-thirds of the population has a 12-times larger OHWF [17] and little predicted growth [18].

HDI [19], was a good predictor of dentist density; and, to a lesser extent, average years of schooling. Whilst there is also evidence that HDI is a predictor for NCD incidence [20,21,22], our findings support the view that it is a health workforce predictor [23]. Despite rapid expansion of higher education in recent years, shortages of high-skilled workers within certain fields, such as medicine, engineering, and dentistry, are a harsh reality [5,15,27]. It has been suggested that further expansion of higher education may need to be more selective to ensure a better match with labour market needs [27], but this is not the only solution as discussed below.

The global influence of traditional American and European models of oral and dental care, which focus on training dentists, has been significant. Despite the philosophical barrier these models may inflict on the development of the OHWF in low- and middle-income countries, several African states have started initiatives to shift the paradigm with successful examples of expansion and use of mid-level providers [7,28,29,30,31]. In line with the UHC targets [32], there is an urgent commitment to focus on preventative actions and social determinants of health to address oral health needs, as oral diseases are preventable in nature. For this, strong leadership is paramount. Higher education funding continues to be a challenge in sub-Saharan Africa despite recent growth [27]. Where evidence exists, it suggests shortages of high-skilled workers within certain fields of study, such as healthcare, may coexist with the rapid expansion of the sector, reflecting mismatches with the labour market [5].

Achieving UHC requires an appropriately educated and trained workforce with a relevant skill mix across community and primary care. A recent technical paper suggests that a seven-fold increase in dentists and dental assistants/therapists is required in the AFR to achieve 70% UHC [32]. Given the enormity of the task, the cost of training dentists and the need to develop a supportive workforce that will work in rural settings, where an estimated 59% of the population reside [24], capacity building should involve shifting from a dentist-centred model towards a diverse, yet integrated, approach [5,8,26]. Interestingly whilst certain Western countries have embraced a wider OHWF skill mix [33,34,35], the AFR is ahead with almost a 1:1 ratio. The role of mid-level providers in delivering healthcare generally has been advocated for the AFR [5], embraced by countries such as Rwanda [29,30], and being explored in Sierra Leone [7,31], Tanzania [36], and Malawi [37] in line with workforce research and guidance [11,38]. Integrating oral health within AFR primary healthcare (PHC) [5,8], whereby community health workers deliver most essential basic oral healthcare [5], with the latter acting as entry points to the health system, will clearly be vitally important for the future. The new WHO global competency and outcomes framework [39] for pre-service health professional training will thus be an important tool. It supports the building of a cadre of community health workers that can promote health, prevent disease, and be the gatekeepers to health services. The Regional Office for Africa has been developing competency-based e-training on oral health for community health workers [40]. In line with this approach, a recent oral health cascade training programme in Malawi [28], delivered by dental therapists trained with international support, has led to oral health promotion by community health workers in rural settings. As mentioned in the FDI World Dental Federation Vision 2030 [41], this is a prime example of co-development with upstream inter-governmental agreement, and a helpful model for supporting or facilitating future change to a more sustainable workforce in Africa [28,41]. Establishing indispensable health worker competencies to guide competency-based education [39,42] supports UHC.

Thus, innovative approaches to education and training that focus on community needs and evidence-informed care pathways, whereby care can be flexibly provided in an integrated manner [43], are essential to tackle the current crisis in the healthcare workforce. Ellard et al., suggest that whilst enhancing knowledge, practical skills, and clinical leadership of mid-level providers in the role of ‘associate clinicians’ may have a positive impact on health outcomes, ‘impact may be confounded by the significant challenges in delivering a service in terms of resources’ [44]; thus, ensuring appropriate resources as highlighted by the response of member states is important for workforce retention.

One of the main strengths of this study was the good response of member states to the survey which, together with the NHWA, provided good data for the AFR. This may possibly be explained by the importance of addressing this critical issue for responding states. However, despite achievements in terms of highlighting OHWF capacity, the focus remains on dentists, and there may be underreporting of mid-level personnel. Greater emphasis should be placed on the wider members of the OHWF team, and this is generally only possible when registration to practise is required, as most will not be employed in the state sector.

This work had the following limitations which should be recognised. First, for some countries, the latest available data on dental assistants and therapists and dental prosthetic technicians are over five years old. Second, not every country provided information on auxiliary members, and the absence of data does not mean that type of workforce is not available or provide insight to their scope of practice. Third, only external validation of data [45] was employed during the development of this research. Fourth, there is no evidence on the number of unregistered OHWF members and no data on the distribution of the workforce across rural and urban settings. Fifth, there is no consistent data on the number and structure of training institutions such as dental schools and training colleges. Sixth, there is no evidence on the working hours for the OHWF (part-time or full-time workers). Seventh, there is no evidence on coverage within a country given that in many settings, the OHWF is concentrated in urban areas, particularly the capital city [7]. Eighth, and finally, given the time lag in reporting data to the NHWA, obtaining contemporary data remains a challenge, particularly in this para-COVID environment.

Nonetheless, the research team created the most comprehensive dataset on the OHWF in Africa; data which need to be kept up-to-date and extended. The immense challenges of the COVID-19 pandemic have deepened oral health inequalities amongst the poorest and most vulnerable populations [46]. Healthcare institutions around the globe are exploring more efficient ways to deliver oral healthcare to people in need [47]. Countries should further strengthen capacity for OHWF data collection and reporting in support of health and planning [15,17,48]. Grouping all mid-level oral health professionals with distinct scope of practice under the umbrella of ‘dental assistants and therapists’ is not particularly helpful, especially for the African context, given variations in the scope of practice amongst mid-level providers [38] of oral health, such as dental assistants, dental nurses, dental hygienists, and dental therapists [30,35,37,49,50]. The OHWF in the AFR is very scarce [5,8]; therefore, being able to plan their distribution and training with greater precision and tailoring their development according to the location needs would be an important step to minimise the burden of oral health diseases of the region. More granularity for each workforce category and annual updates through NHWFA will be important to strengthen data quality and assist strategic planning.

In summary, there is an urgent need to develop primary care services with an appropriate workforce to address the pain, suffering, and even death from acute dental conditions [1,5,7]. Given the size of the challenge, there is great scope for innovative models of care [51,52] within a public health approach involving upstream action. Workforce capacity building should be contextually appropriate to deliver UHC in a sustainable manner rather than merely following traditional dentist-focused approaches. Greater collaboration has been strongly recommended to support capacity building [15,39,53], ideally working in partnership (involving co-production) [11,28,31,52] with successful examples emerging [29,36,37]. African countries may explore working together creatively to rethink the shape of health systems specific to their current needs and share learning. The recent publication of country profiles as part of the Global Oral Health Status Report provides important information to support all nations in preparing for effective action [54].

## 5. Conclusions

There is a stark oral health workforce deficit in the AFR; thus, there is an urgent need for developmental action supported by capacity building to address oral health needs, with great potential to develop innovative models of care which further utilise the workforce skill mix and community health personnel. This should be supported by oral health policy and health systems strengthening in support of achieving UHC.

## Figures and Tables

**Figure 1 ijerph-20-02328-f001:**
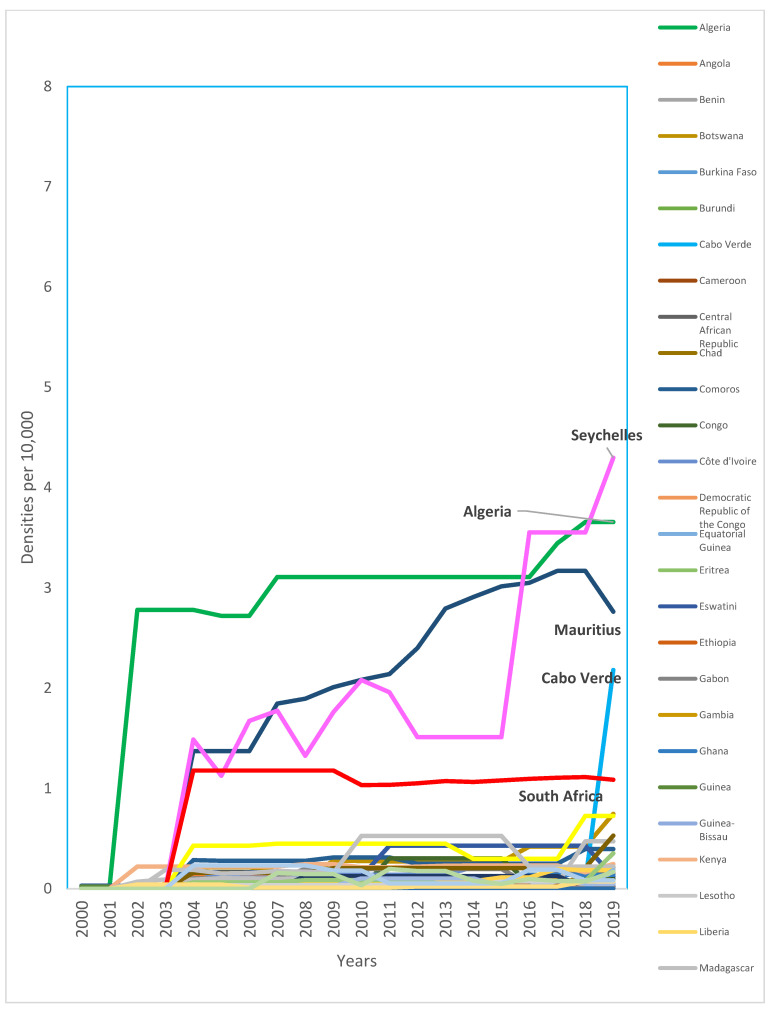
Trends in density of dentists per 10,000 population for the AFR, 2000-2019 (Appendix D). Source: The workforce data are based on the latest available data in the NHWA data platform as of 31 March 2021, apart from the data for 2019, which used a combination of the latest available data from the NHWA data platform and King’s College London survey. Countries with the highest densities were named in the plot and highlighted in bold.

**Figure 2 ijerph-20-02328-f002:**
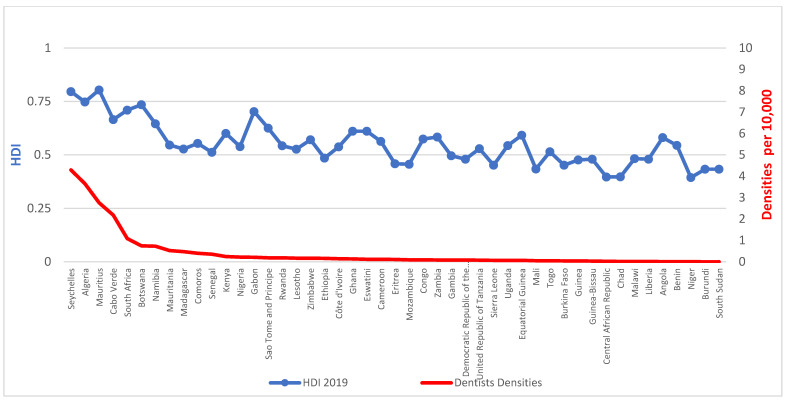
Densities of dentists per 10,000 population and Human Development Index (HDI) for the AFR, 2019. Source: The workforce data are based on a combination of the latest available data from the NHWA data platform and King’s College London survey. HDI was collected from the latest Human Development Report from the United Nations Development Programme (UNDP).

**Figure 3 ijerph-20-02328-f003:**
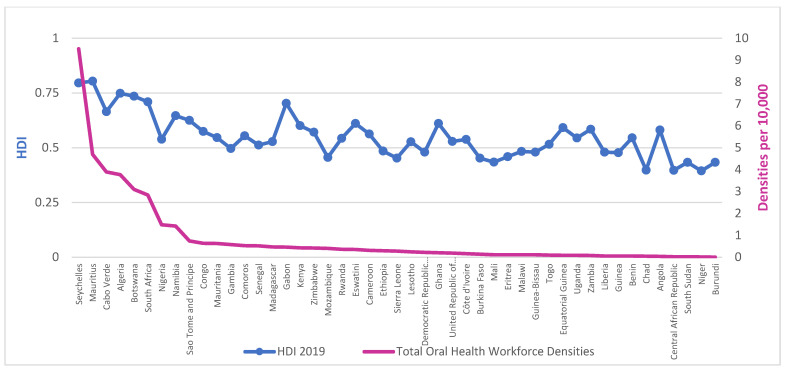
Densities of combined OHWF per 10,000 population and Human Development Index (HDI) for the AFR, 2019. Source: The workforce data are based on a combination of the latest available data from the NHWA data platform and King’s College London survey. HDI was collected from the latest Human Development Report from the United Nations Development Programme (UNDP).

**Figure 4 ijerph-20-02328-f004:**
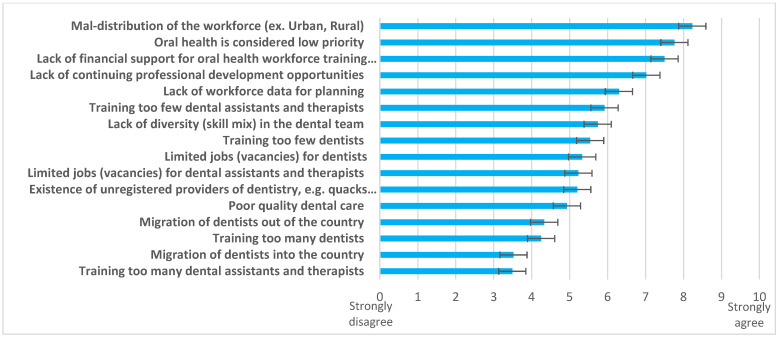
Challenges related to the OHWF highlighted by the AFR, 2019. Source: King’s College London survey, 2019.

**Figure 5 ijerph-20-02328-f005:**
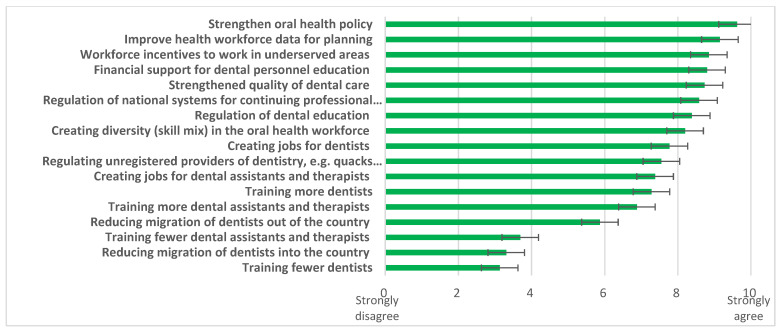
Possible solutions related to the OHWF highlighted by the AFR, 2019. Source: King’s College London & WHO Survey, 2019.

**Table 1 ijerph-20-02328-t001:** Unadjusted linear regression analysis of dentists per 10,000 population in the AFR, 2019.

	Coefficient	Sig.	95% Conf. Interval
Human Development Index [HDI]	6.534	<0.001 *	4.670, 8.399
Mean Years of Schooling	0.231	<0.001 *	0.124, 0.337
Number of dental schools	0.050	0.522	−0.108, 0.209
% Urban population	0.009	0.217	−0.005, 0.022

* Statistically significant. Source: The workforce data are based on the latest available data in the NHWA data platform as of 31 March 2021, apart from the data for 2019, which uses a combination of the latest available data from the NHWA data platform and King’s College London survey. Linear regression analysis was performed in the Statistical Package for the Social Sciences (SPSS) 27 version.

**Table 2 ijerph-20-02328-t002:** Unadjusted linear regression analysis of combined OHWF per 10,000 population in the AFR, 2019.

	Coefficient	Sig.	95% Conf. Interval
Human Development Index [HDI]	12.072	<0.001 *	8.588, 15.577
Mean Years of Schooling	0.460	<0.001 *	0.268, 0.652
Number of dental schools	0.022	0.880	−0.274, 0.318
% Urban population	0.016	0.210	−0.010, 0.042

* Statistically significant. Source: The workforce data are based on the latest available data in the NHWA data platform as of 31 March 2021, apart from the data for 2019, which uses a combination of the latest available data from the NHWA data platform and King’s College London survey. Linear regression analysis was performed in the Statistical Package for the Social Sciences (SPSS) 27 version.

## Data Availability

Data available in a publicly accessible repository that does not issue DOIs. Publicly available datasets were analyzed in this study https://apps.who.int/nhwaportal/Home/Index (accessed on 10 March 2021). Data is contained within the article or in the Appendix A.

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
