# Peer review of "Oral Health Workforce in Africa: A Scarce Resource"

_ijerph, 2023, doi:10.3390/ijerph20032328_

Round 1
Reviewer 1 Report
Authors provide a comprehensive evaluation of the contemporary oral health workforce in the African region. It highlights the urgent need to strengthen policy, health, the education systems to expand the workforce and meet the need of the African region.
Minor
1. The color code used in Fig. 1 is indistinguishable. Please provide a table of the corresponding data in supplemental data.
2. Supplemental materials are not cited in the manuscript. However, there is a section named "Supplemental Materials". And Figure S1, Table S1, Video S1 are listed but not available. Please check.
3. The alignment of row 4 in table 1 is not consistent with others. Please correct.
Author Response
- The colour code used in Fig. 1 is indistinguishable. Please provide a table of the corresponding data in supplemental data.
- To assist with reading Figure 1, Appendix D has been created and added to the supplementary materials. This new table provides the corresponding data used in Figure 1.
- Supplemental materials are not cited in the manuscript. However, there is a section named "Supplemental Materials". And Figure S1, Table S1, Video S1 are listed but not available. Please check.
- All supplemental materials (appendices) cited in our manuscript have been included in the ‘Supplementary Material’ document. We have renamed the file to make this clear.
- The alignment of row 4 in table 1 is not consistent with others. Please correct.
- This has been corrected, thank you.
Please see supplementary material attached.

Reviewer 2 Report
Review on Gallagher et al’s Oral Health Workforce in Afrika: a scarce resource
The study provides new information on the dental workforce availability and distribution in Africa. It is well written. Limitations are also described in the study. In the discussion there are suggestions for how to solve the current situation in Africa. The lack of money, schools, trainers at some countries the presence of civil war (Subsaharan region, Ethiopia, .etc) will not make the situation easier. Do the authors think that the WHO can take a more pronounced role in the development of dental education in Africa?
Author Response
- The study provides new information on the dental workforce availability and distribution in Africa. It is well written. Limitations are also described in the study. In the discussion there are suggestions for how to solve the current situation in Africa. The lack of money, schools, trainers at some countries the presence of civil war (Sub-Saharan region, Ethiopia, etc) will not make the situation easier. Do the authors think that the WHO can take a more pronounced role in the development of dental education in Africa?
- Absolutely, we agree with reviewer 2 and believe that the WHO is already working on this. In our Discussion section, between rows 286 and 298, we have provided examples of how the WHO has been supporting oral health workforce development through educational initiatives. Also, the recent publication of the Global Oral Health Status Report (reference 54) provides important information to support all nations in preparing for effective action.
Reviewer 3 Report
The study assesses the distribution and density of oral workforce in the African region, according to the
AFR Regional Oral Health Strategy 2016-2025 wich aims to integrate oral health into NCD-prevention and control, in the context of Universal Health Coverage.
Oral diseases have been recognized as very important in the context of NCDs and the burden of oral diseases has been recognised by the WHO and the need to create workforce models to deal with those diseases is imperative.
Data regarding the distribution and density of the workforce in this particullar region is scarce and this study shows a very broad description of the whole area, creating baseline data to further devolep the models as requested by the WHO.
There are some limitations of the study, starting with the fact that is has been conducted over 2 years ago, but the authors addressed those limitations and explained that the study still represents a comprehensive evaluation of the regional data.
The conclusions are consistent and address the objectives.
The references are appropiate and up to date.
The tables are fine. Also the supplementary material is clearly presented. Maybe the first two figures could be made more clear. There are a lot of countries presented in the graph and it is rather unclear how to interpret them.
Author Response
The tables are fine. Also the supplementary material is clearly presented. Maybe the first two figures could be made clearer. There are a lot of countries presented in the graph and it is rather unclear how to interpret them.
- As outlined above in response to Reviewer 1, Appendix D has been created in the supplementary material document to present a table with all corresponding data used in Figure 1.
Please see attachment.
